# “In the Village That She Comes from, Most of the People Don’t Know Anything about Cervical Cancer”: A Health Systems Appraisal of Cervical Cancer Prevention Services in Tanzania

**DOI:** 10.3390/ijerph21081059

**Published:** 2024-08-13

**Authors:** Melinda Chelva, Sanchit Kaushal, Nicola West, Erica Erwin, Safina Yuma, Jessica Sleeth, Khadija I. Yahya-Malima, Donna Shelley, Isabelle Risso-Gill, Karen Yeates

**Affiliations:** 1Department of Medicine, Queen’s University, Kingston, ON K7L 3N6, Canada; 2Temerty Faculty of Medicine, University of Toronto, Toronto, ON M5S 1A8, Canada; 3Ministry of Health, Community Development, Gender, Elderly and Children, Dodoma P.O. Box 743, Tanzania; 4Canadian Cancer Trials Group, Kingston, ON K7L 2V5, Canada; 5Department of Nursing Management, Muhimbili University of Health and Allied Sciences, Dar es Salaam P.O. Box 65001, Tanzania; 6School of Global Public Health, New York University, New York, NY 10003, USA; 7Specialty Registrar in Public Health, National Health Service (NHS), London SE1 8UG, UK

**Keywords:** cervical cancer, knowledge, education, beliefs, health systems appraisal, Tanzania

## Abstract

Introduction: Cervical cancer is the fourth most common cancer in women globally. It is the most common cancer in Tanzania, resulting in about 9772 new cases and 6695 deaths each year. Research has shown an association between low levels of risk perception and knowledge of the prevention, risks, signs, etiology, and treatment of cervical cancer and low screening uptake, as contributing to high rates of cervical cancer-related mortality. However, there is scant literature on the perspectives of a wider group of stakeholders (e.g., policymakers, healthcare providers (HCPs), and women at risk), especially those living in rural and semi-rural settings. The main objective of this study is to understand knowledge and perspectives on cervical cancer risk and screening among these populations. Methods: We adapted Risso-Gill and colleagues’ framework for a Health Systems Appraisal (HSA), to identify HCPs’ perspective of the extent to which health system requirements for effective cervical cancer screening, prevention, and control are in place in Tanzania. We adapted interview topic guides for cervical cancer screening using the HSA framework approach. Study participants (69 in total) were interviewed between 2014 and 2018—participants included key stakeholders, HCPs, and women at risk for cervical cancer. The data were analyzed using reflexive thematic analysis methodology. Results: Seven themes emerged from our analysis of semi-structured interviews and focus groups: (1) knowledge of the role of screening and preventive care/services (e.g., prevention, risks, signs, etiology, and treatment), (2) training and knowledge of HCPs, (3) knowledge of cervical cancer screening among women at risk, (4) beliefs about cervical cancer screening, (5) role of traditional medicine, (6) risk factors, and (7) symptoms and signs. Conclusions: Our results demonstrate that there is a low level of knowledge of the role of screening and preventive services among stakeholders, HCPs, and women living in rural and semi-rural locations in Tanzania. There is a critical need to implement more initiatives and programs to increase the uptake of screening and related services and allow women to make more informed decisions on their health.

## 1. Introduction

Cervical cancer is the fourth most common cancer in women globally [1]. Unfortunately, cervical cancer primarily affects medically underserved women and is disproportionately higher in low- and lower-middle-income countries, accounting for 85% of the cases worldwide and 90% of deaths [1]. While there has been a reduction in cervical cancer prevalence rates since the introduction of Pap smears in the mid-20th century, healthcare systems and care vary across the globe [2]. This trend was not observed in low- and lower-middle-income countries (LMICs), due to a lack of health and human resources for screening implementation and low population coverage [2].

In 2020, The WHO launched a global strategy to accelerate the elimination of cervical cancer by 2030 [3]. The goals include (1) vaccinating 90% of eligible girls against Human Papillomavirus (HPV); (2) screening 70% of eligible women at least twice in their lifetimes (ideally at age 35 and 45 years); and (3) effectively treating 90% of those with a positive screening test or a cervical lesion, including palliative care when needed [3,4,5]. Presently, the WHO recommends using HPV DNA sampling as the primary screening test as opposed to visual inspection with/through acetic acid (VIA) or cytology for screening and treatment purposes among both the general population, as well as women living with HIV [4,5,6]. Tanzania plans to begin the implementation of HPV testing as a primary screening modality in a pilot program in 2024, but VIA will remain the primary screening modality in the national cervical cancer prevention program until this program can be fully brought to scale nationally [6,7,8,9].

Notably, cervical cancer has been identified as the most common cancer among women in Tanzania, resulting in about 9772 new cases and 6695 deaths each year [2]. This is especially alarming, given that over 60% of the population is under the age of 25 [2]. Unfortunately, insufficient surgical equipment, medications, and imaging capabilities are only a few of the reasons that explain this statistic [2]. Research has identified that a lack of awareness and knowledge of cervical cancer and available screening methods is a major contributor to the low screening uptake [2]. Studies in Tanzania suggest several barriers to screening uptake among women at risk and these include knowledge of cervical cancer and related risk and the role of screening in prevention, knowledge among healthcare providers (HCPs) regarding cervical cancer and prevention, and misconceptions and beliefs regarding cervical cancer and the role of screening [2]. In addition, financial barriers for women who undergo screening and have access to preventive services at a higher level of care or travel long distances to receive these services continue to exist [2]. Many women have received education on cervical cancer and the role of screening in secondary prevention of cervical cancer and/or the role of HPV vaccination as primary prevention (i.e., among younger women) through radio and television awareness campaigns [10]. Among these women, one-third knew someone who suffered from cervical cancer [10]. Notably, predictors of awareness of cervical cancer and the role of screening included having a secondary education or higher, having 1–4 children, being 30–44 years old, living in an urban household, and obtaining a reasonable income [10]. Despite urban status being a predictor of awareness, outcomes from a community-based cross-sectional study conducted in five municipalities of Dar es Salaam, the largest urban center in Tanzania, found that the three predictors of cervical cancer screening were knowledge level, screening intention, and health beliefs [11]. It was found that 53% of participants had inadequate knowledge of cervical cancer and cervical cancer screening, with 91.8% of the overall population having never previously been screened for cervical cancer [11].

While prior studies on the effects of knowledge levels on cervical cancer screening uptake in rural and semi-rural Tanzania have been explored, there is scant literature on the perspectives of not only women at risk, but also other health system stakeholders (e.g., policymakers, program managers, and HCPs) [12]. Several studies have suggested that many women, particularly those with low levels of knowledge about cervical cancer and screening, may not recognize the benefit of screening over the possible consequences of forgoing screening [12]. In one study, it was determined that while the majority of the women had heard of cervical cancer, only 6–21% had ever been screened [2]. Interestingly, women from more developed regions in Tanzania were more likely to be screened if they had some form of health insurance, were older, or had some prior knowledge of cervical cancer [2]. In addition, across seven studies that have been conducted regarding cervical cancer prevention among Tanzanian women, it was identified that 80% of women had heard of cervical cancer, although knowledge levels surrounding specific prevention methods and risk factors were low [2]. Lack of knowledge and awareness regarding cervical cancer and the role of screening and preventive services extends beyond women at risk, with prior studies demonstrating a lack of HCP knowledge regarding cervical cancer in the region [13]. For example, one study found that, before a 1-day cancer training symposium conducted amongst 43 clinicians, 69% had reported never receiving any training regarding cervical cancer and cervical cancer screening in the past [13]. The main objective of the study is to understand the perspectives that policymakers, HCPs, and women at risk have towards cervical cancer and screening services, especially among those living in rural and semi-rural settings.

## 2. Methods

Given the existing evidence, we designed and conducted a cross-sectional research study using semi-structured interviews. Most HCPs and women interviewed resided and worked in rural or semi-rural locations in the Northern Zone of Tanzania (Arusha, Kilimanjaro, and Tanga Regions). For representation from the national screening program and national cancer experts, we also sought key informants to participate in interviews at the National Cancer Hospital in Dar es Salaam, (Ocean Road Cancer Institute (ORCI)). We developed our methodology from the HSA framework that had been previously utilized in LMICs for assessing multi-level health system barriers and enablers for the screening, diagnosis, and management of chronic diseases (such as hypertension and diabetes) [14,15].

### 2.1. Theoretical Framework

The approach to the health systems assessment is described in Figure 1 (below) and underpinned by a conceptual framework developed by Risso-Gill and colleagues [14,15]. The framework incorporates the WHO health system building blocks and recognizes that the building blocks (leadership and governance, service delivery, health system financing, health workforce, medical products, vaccines and technologies, and health information systems) exist in a complex ‘system’ and different components may influence outcomes of another and allows identification of how resource inputs may influence outcomes in another [16].

### 2.2. The Cervical Cancer Prevention Program of Tanzania

Tanzania has a “multi-tiered, comprehensive cervical cancer control programme” [17]. Figure 2 demonstrates components embedded across Tanzania’s screening and treatment pathway. There has been significant progress in cervical cancer screening progress. Since 2010, the government has established a national Cervical Cancer Prevention (CECAP) program offering VIA through a single-visit approach as a national strategy [18,19]. Tanzania’s CECAP program includes a mHealth-supported quality assurance program developed by Dr. Karen Yeates and colleagues (formerly known as the ‘SEVIA’ application) [18,19]. Eligible women are screened for cervical cancer using VIA [20]. VIA is free, and treatment of pre-cancerous lesions (e.g., thermal ablation and LEEP) is at no cost through the program [20]. Unfortunately, there is a low cryotherapy rate, as a result of a lack of resources and insufficient equipment [20]. Notably, there are costs involved for diagnostic services [20]. Comprehensive services that include diagnostic, radiotherapy, chemotherapy, and palliative care are offered in two public sector institutions [20]. There are costs for further assessment and higher levels of treatment (e.g., cervical biopsy and surgery) [20].

This research study is part of a series of studies evaluating HCP, women at risk, and other stakeholder perspectives on determinants of the cervical cancer screening and preventive treatment program in Tanzania.

We adapted the HSA framework, conceptualized by Risso-Gill, Balabanova, and colleagues for hypertension control (Figure 1) [14,15]. The health systems assessment has been leveraged in other low-resource settings (including Malaysia and Columbia) to understand the health system barriers to hypertension diagnosis and management [14,15]. While cervical cancer screening and prevention services are dissimilar to the diagnosis and management of hypertension, the HSA framework captures multiple levels of health system components and inputs required for screening, diagnosis, and management of almost any chronic, non-communicable disease [14,15]. This allows for ease of adaptation and significant overlap with the concept of cervical cancer screening and the ‘care pathway’ for cervical cancer prevention services and required follow-up in almost any healthcare system [14,15]. Therefore, we used the foundations of this model/framework to inform our research approach and evaluate policymaker, health system leadership, and patient and HCPs’ perspectives of the extent to which health system requirements for effective cervical cancer screening, prevention, and control are in place in Tanzania [14,15]. The framework posits that a coordinated investment that includes physical resources (e.g., diagnostic equipment), human resources (e.g., trained healthcare staff), intellectual resources (e.g., evidence-based guidelines), and social resources (e.g., patient support systems) are necessary inputs for achieving clinical outcomes [14,15]. Further, additional contextual factors (e.g., attitudes and beliefs among providers and women at risk) may modify the impact of those inputs on the implementation process [14,15]. The inputs and context will be evaluated from the perspective of women at risk (patients) and HCPs and health system stakeholders (in our case, cervical cancer screening providers, program managers, policymakers, and other health providers who perform roles in cervical cancer prevention services) to better understand the full range of barriers and facilitators regarding cervical cancer screening uptake and access to preventive services in Tanzania.

Using previous studies as a guide, the HSA framework informed the design of this study, providing a structured process by which to identify critical strengths and weaknesses in the management of cervical cancer screening and preventive treatment services in Tanzania and, subsequently, identify feasible strategies that offer the possibility of improving or optimizing cervical cancer control at a national level while also considering service delivery and uptake among women at risk who reside in rural or semi-rural locations [14,15]. The data were analyzed using thematic analysis methodology [21,22].

We adapted interview topic guides for cervical cancer screening and preventive treatment services using the HSA framework [14,15]. The following are the main themes for each of the interviews that were explored: (1) pathways to care, (2) access to care, (3) cervical cancer control, (4) physical resources, (5) financing, (6) patient communication and empowerment, (7) human resources, (8) knowledge and information resources, (9), policy and governance, and (10) cervical cancer screening strategies.

### 2.3. Study Participants

Fieldwork was undertaken by local researcher team members who received training in health systems appraisal at a workshop held in Moshi, Tanzania, using a detailed manual (HSA Toolkit) adapted from the one developed for hypertension that contained sampling frameworks and assessment tools to better standardize the participant recruitment and the interviews and data collection across sites. At each site, researchers identified the main health facilities where women would seek cervical cancer screening and treatment services in the national screening program.

Study participant categories and recruitment are displayed below in Table 1. The eligibility criteria and recruitment of participants were completed through consultation with stakeholders (e.g., Tanzanian Ministry of Health officials). The inclusion criteria included women who had undergone cervical cancer screening, and HCPs who had provided cervical cancer screening or who had been involved in cervical cancer screening. The exclusion criteria included women who had never been screened or providers who had never conducted screening. Enrolment was performed opportunistically (e.g., the study team visited health facilities that had cervical cancer screening sites that were supported by the national Cervical Cancer Prevention Program to invite participants for interviews with the consideration of sampling across various roles and cadres of healthcare, and specific sampling of rural and semi-rural sites where possible. This included cervical cancer screening program leadership (rural/urban; private/public) whose work is related to the provision of cervical cancer screening and preventive treatment services and women at risk (reflective of age and HIV status). Key stakeholders were divided into four main categories: program leadership/policymakers, HCPs, women at risk, and non-physician HCPs. We recruited 8 program leadership/policymaker participants that comprised national- and district-level government officials, donor agency leaders, non-governmental organizations, health facility managers, pharmaceutical company representatives, and academia (who know cervical cancer screening and cancer care). We recruited 15 HCPs from primary care/outpatient clinics and village dispensaries where cervical cancer screening may or may not be provided, secondary (health centers) and tertiary (hospitals), health facility-based medical doctors, and clinical officers. We recruited 9 non-physician HCPs including matrons, nurses, and nurse assistants. Women at risk of cervical cancer (women at risk/clients) included 39 women (PA) from various districts, from screening sites at differing health system levels including up to the national cancer hospital. A total of 71 Interviews were conducted at 11 sites between 2014 and 2018 (Mt. Meru: 2 HCPs and 1 PA; Marangu: 2 HCPs and 3 PAs; Meru: 2 HCPs and 3 PAs; Pasua: 1 HCP; TPC: 4 HCPs and 5 PAs; KCMC: 2 KIs and 4 PAs; Kibosho: 1 HCP and 3 PAs; ORCI: 6 HCPs, 3 KIs, and 8 PAs; Kilema: 3 HCPs; Pamoja: 6 PAs; Bombo: 1 KI). A large proportion of the HCPs and women at risk are from rural or semi-rural locations (Table 2).

## 3. Data Collection

Semi-structured individual interviews were conducted using an interview guide derived from the Health Systems Appraisal Toolkit provided to our research team by Risso-Gill. All the participants agreed to be interviewed. Trained research students and research assistants completed the interviews. The research students were given training using the material provided by the London School of Hygiene and Tropical Medicine. The research students were all accompanied by a trained Tanzanian research assistant. Two longer-term staff initiated the HSA process including networking/communication with stakeholders and obtaining permissions. Some interviews were conducted by them, and some were conducted by the research assistants (who received training in conducting an HSA). A purposive sampling technique was used to recruit key stakeholders involved in cervical cancer screening program leadership and policy (including policymakers and HCPs) and to ensure a range of experiences regarding optimal management and control of cervical cancer within Tanzania were captured. In addition, women at risk of cervical cancer were recruited for semi-structured interviews to share their own perspectives and knowledge on cervical cancer and cervical cancer prevention services in Tanzania. An open-ended semi-structured interview guide, which included items that explored the availability of physical resources, human resources, intellectual resources, and social resources necessary for optimal management and control of cervical cancer was utilized. The open-ended questions allowed participants to elaborate on issues they considered important or relevant, particularly about these categories. All participants described their role and position within the health systems and in the context of cervical cancer screening and preventive treatment/management. Participants shared their perceptions on how women at risk are diagnosed with cervical precancer or invasive cervical cancer within the healthcare system and the pathways for preventive treatment, referrals for required management, preventive follow-up care, access to cervical cancer treatment, lifestyle counseling, and support with patient-centered care for cervical cancer control (Figure 3). All interviews were audio recorded with written consent from participants. Interviews lasted between 30 and 45 min and were conducted in a health facility.

### Data Analysis

We analyzed the data using an iterative process. Transcripts from interviews were transcribed verbatim and translated (if conducted in Kiswahili) and exported to NVivo for Mac v11.4.0 (QSR International, Melbourne, Australia) for analysis, where two of the authors independently became familiar with and reviewed the data in depth. A deductive approach was utilized based on the health systems assessment framework by Risso-Gill and colleagues. Statements related to physical, human, intellectual, and social resources were coded as their own categories, and a detailed thematic analysis was performed to extract the salient themes. An open coding approach was adopted, forming a general description of the research topic by generating categories and subcategories as they emerged. This systematic approach was appropriate for open-ended interviews to determine trends and patterns. Upon completion of individual coding, the two coders (MC and SK) convened on multiple occasions to discuss the coding process and explore emerging themes. This was performed until coding saturation was achieved, and discrepancies in coding were discussed until consensus was reached. Quotes reported in the results represent statements that are highly representative of each theme and are cited as indicated by key stakeholders (cervical cancer screening program leadership), HCPs, non-physician HCPs, and women at risk.

## 4. Results

Thematic analysis based on initial coding and development of a codebook was used to identify emergent themes and triangulate the information collected, facilitated by NVivo for Mac v11.4.0 (QSR International, Melbourne, Australia). This inductive approach allowed for links between research objectives and findings from raw data (Figure 1). It helped develop a deeper understanding of emergent themes’ underlying experiences and processes. Verbatim quotations were frequently used to illustrate the responses of the respondents on important themes and sub-themes. Seven themes emerged from our analysis: knowledge of the role of screening and preventive care/services (e.g., prevention, risks, signs, etiology, and treatment), training and knowledge of HCPs, knowledge of women at risk, beliefs about cervical cancer screening, traditional medicine, risk factors, and symptoms and signs (Table 3).

The first theme investigated awareness and knowledge levels among women at risk and HCPs regarding cervical cancer and the role of preventive services. Sub-themes included the importance of health promotion, methods of raising awareness such as word of mouth and radio, and underlying reasons behind the refusal of screening. The second theme was the education of HCPs. A relevant sub-theme was the relation of training to the capability and knowledge that HCPs possessed. The third theme was the education level of women at risk and opportunities for improving awareness through education. The fourth theme that emerged was a belief in cervical cancer screening among women and HCPs. The fifth major theme addressed traditional healers and traditional medicine and its relevance to cervical cancer screening amongst the population. The sixth theme addressed common risk factors related to cervical cancer. Sub-themes that emerged in the analysis include HIV and early sexual intercourse. Finally, the seventh theme discussed the signs and symptoms women at risk experience with cervical cancer and/or HCPs commonly identify. Pain was identified as a sub-theme of interest.

Knowledge of the role of screening and preventive care/services

Many women at risk expressed a lack of awareness surrounding cervical cancer prevention, risks, signs/symptoms, etiology, and treatment, and overall, the importance of screening.


*“Women are not so much aware of cervical cancer so it will take time for them to become aware about understand the importance of screening for their health. I just educate them and then it will take time for them to think and to prepare themselves, and then they will come—so it takes time”.*


The need for more education surrounding the disease and prevention was highlighted.


*“So, she said that awareness should be spread in different areas. Because she mentioned the village that she comes from, most of the people don’t know anything about cervical cancer. So, awareness should be provided”.*


Health Promotion

Interviewers asked HCPs if there was a reason that women did not come in for screening. The HCP compared health promotion for HIV vs. cervical cancer. They discussed the need to implement more initiatives that will educate women on the disease and the need to be screened. Other factors, such as transportation and distance were identified as additional barriers to health promotion.


*“So, it just started, and people are not that well informed. And the project, is not that well, not that well organized, compared to maybe ctc [HIV clinic] when we can go to the villages and talk to people about HIV, reach out to them. For cervical cancer it is different, we don’t have such activities going on, or such organized procedures or activities. For maybe a certain period you have screened a certain number of people or go to the villages to talk to them so they can come to be screened—they are not that established. So, you’ve got to get a larger number of people and also difficult to get the information to the people if you are not going to them. Because they won’t come here, you have to go to them. It is different in the town, where people are well informed about their health, and they have to follow. But in the villages, you have to follow them—tell them, teach them”.*


Word of Mouth

Word of mouth was identified as the primary means of spreading awareness. Once women attended their screening visit, they would often share their experiences with their family, friends, neighbors, and community.


*“The ones who have already come are the ones who will go back and tell like she was told by the neighbor that it’s fine, there’s no problem. So, when they go back and tell them, they will come”.*



*“She says that not everyone knows, but a lot of them know. Mostly because if they come here then they’re told about it, and then those who go back actually tell their colleagues about it. She’s even been told by some of her colleagues that she should bring them here”.*


Radio

Awareness and information surrounding cervical cancer are largely spread through announcements made on the radio or TV.


*“From the radio, I heard that cervical cancer is a dangerous disease, so it is good to do screening at the earlier stage. So that is why I decided to go to the clinic for the check-up”.*



*“Almost everyone in this country, in Africa has a radio. Or radio access. So, you cannot be scared they’ll get your information- then comes, of course, the TV, which is not all that, not everyone has a TV, but many people have TVs”.*


Refusal of Screening

Some women who are already at the clinic refuse to be screened and receive treatment, due to fear or lack of understanding about the role of screening and treatment for cervical cancer prevention.


*“Of course, yea challenges—they may refuse treatment like cryotherapy—like one client refused cryotherapy.—and others when you refer them to KCMC don’t go. But most they agree to be screened”.*



*“Most women, accept this screening only a few have not already decided about screening. From today, only three women refused to do the screening, so I’m not sure if they are going to accept or they will just leave”.*


Another woman described her thoughts on the refusal of screening:


*“Some they are afraid, ignorance..... we are going into the church, we are giving information to the church. Even to the market, the schools, but others are not coming”.*


2.Education (HCPs)

A lack of education surrounding cervical cancer exists at the HCP level. The level of knowledge is often variable among HCPs of different ages, experience levels, and professional levels.


*“I don’t think even among the health profession, that, I don’t know which percentage I should mention. But I don’t think that people know quite—that people know exactly how to walk through it”.*



*“Updating, more education to the community and also training more staff to know about cervical cancer”.*



*“So, you have these young, young doctors, only the old doctors don’t know, old nurses who have trained twenty years ago, they are probably not very aware about cervical cancer. They are aware of cervical cancer but, screening and a lot of information about it, no. In terms of patients, the population I don’t think they have a lot of information on cervical cancer, no”.*


Training

HCPs highlighted the need for HCPs across all levels to be trained to conduct cervical cancer screening to improve reach and uptake.


*“They should educate more people on the job training we’re two…. when we will train more staff here our services will improve….and in the rural areas we need to train more staff and home service care providers…. of course, when you talk with them they don’t know so much about the cervical cancer screening”.*



*“I think that we have to train people in different categories, yeah. Train the specialists, train the other nurse- the other doctors, sorry, the other practitioners, the nurses, the advanced medical officers, so that every person can screen”.*


3.Education (Women at Risk)

There is a pressing need for education related to cervical cancer, including the reasons for development and the importance of being screened.


*“They come from their home to here, and we give them the education…. health education……the general issues about cervical cancer and after that for those who are willing to screen their health…. they are screened….and then they get their result if it is positive or negative. If it is negative, we give them the next date for check, but for those whom the problem is so big and we feel we can’t solve, we refer to other places”.*


One woman described how she believes women can be better reached and provided with cervical cancer screening and care:


*“Provision of education is needed. And also, on that education, they are supposed to give so that they can get it even through men. So, if they know that, a lot of women would be willing to go for screening”.*


4.Beliefs on Cervical Cancer Screening

Several misconceptions related to the cause of cervical cancer continue to exist. Some women believed that using contraceptives or consuming certain foods resulted in the development of cervical cancer.


*“She said that this contraception can cause cervical cancer”.*


Another woman described her thoughts on cervical cancer and women’s health.


*“Before I knew about cervical cancer, I knew it’s a disease you can get from different medicines which a lot of women like to use. Or you can get different types of food, which you like to eat. Also, I didn’t know if this disease if you can get it from men. I thought maybe it was only for women. Before I know that”.*


5.Traditional Medicine

A large dependence and belief surrounding the efficacy of traditional medicine continues to exist. A combination of beliefs and financial strains cause many women to use local herbs, instead of chemotherapy to treat their cancer. One HCP mentioned:


*“They want to go to a traditional attendant (inaudible 38:20) or a traditional doctor who can say, this is just someone who has bewitched you, I can take care of you. They can understand”.*


Another HCP explained:


*“One patient came and said if you diagnosed with cervical cancer in KCMC she used the local herbs, but they don’t treat anything?”*



*“The education provided by KCMC helped us, and we got no disturbance and when you go there with your husband, they give you priority. (move that to another theme- treatment?)”.*


One woman discussed the financial burden of chemotherapy:


*“If I go to a traditional doctor and they say, ‘bring on God I can take care of this’ and a god will cost about 150–200,000 Tanzanian shillings. And you go to Ocean Road [National Cancer Hospital of Tanzania]. One course of chemotherapy will cost 400,000. Okay? So which options will you take? If you are my relative looking after me? You going to take the easiest option available. So, that’s still a big gap”.*


6.Risk Factors

HCPs explain the role of HIV in cervical cancer.


*“High risk—patients with- who are HIV positive. And the screening, because there is no formal clinic, we usually do screening based on request patients who have requested, and then we have other satellite screening sites. They refer like this patient has a problem, please take a look at her—we’ll do that”.*


One woman explained her knowledge surrounding HIV.


*“They mentioned about HIV. That the one who has HIV-positive are at risk of getting this cervical cancer”.*


HIV

HCPs highlighted HIV as a significant risk factor for cervical cancer.


*“Yeah, there are so many risk factors. HIV is just one of them and it has been publicized because with HIV the disease progression is very fast, but there are many other risk factors for getting cervical cancer”.*



*“The low immunity, and for those who start sexual intercourse from the beginning are in risk to get the cervical cancer…and also those who get the human papillomaviruses…they are also at risk to get the cervical cancer”.*


Early Sexual Intercourse

Implementing sexual reproductive health classes for young girls may educate women on the importance of using contraceptives and prevent the development of cervical cancer.


*“Yeah, and also another challenge is for young kids. Before they are grown up, sometimes they have a lot of—they have multiple partners, for them, they don’t know it is risky. Or whatever, for cervical cancer”.*



*“We have to intensify this sexual reproductive health to these young ladies so that they can delay sexual intercourse so if we manage to go directly to the school to input information for cervical cancer prevention it will be better than staying until you get suffer then you look for the service”.- use a narrative to explain more”.*


7.Symptoms and Signs

Pain is one of the most common symptoms of cervical cancer.


*“I visited before, I was getting pain in the stomach and then I went to the hospital and then the doctor recommended to me to do cervical cancer screening”.*


Other complications women described included bleeding and discharge.


*“You can see the elder women come to you and complain about the PV bleeding, post-coital bleeding and follow the discharge and when you insert the speculum you see there is cancer”.*


Pain

Common symptoms women reported included abnormal or pelvic pain and bleeding. These were often the first few signs women experienced before going to get screened.


*“Because I have this abnormal stomach frequently, that is why also I decided to do screening”.*


Another woman shared her experience:


*“Symptoms caught me when I was at home it started by getting abdomen pain and when I had sex, I felt pain and blood coming out when I was on my period the period lasted for long like a month, I went to Mawenzi hospital and got medication”.*


## 5. Discussion

The main objective of the study is to understand the perspectives of a wider group of stakeholders (e.g., policymakers, healthcare providers (HCPs), and women at risk) on cervical cancer and screening services, especially those living in rural and semi-rural settings. The findings highlight existing challenges in cervical cancer screening across Tanzania. Rural areas particularly struggle with numerous barriers that limit not only access to cervical cancer screening services, but also access to multiple healthcare services [23]. Lower health literacy and limited knowledge regarding screening services or insufficient infrastructure also play a role [23]. Moreover, cultural beliefs significantly shape health-seeking behavior with respect to cervical cancer screening uptake [23]. Women who reside in urban locations may be more acquainted with where to access specific healthcare services [23]. Addressing these issues requires targeted efforts, especially in rural regions, including the implementation of mobile screening initiatives, enhanced health education campaigns, integration of community health workers or other extenders of the health workforce who can provide health promotion at the community level, and health system infrastructure strengthening and enhancements. These measures are crucial for ensuring equitable access to cervical cancer screening services throughout the country.

The findings of our study intersect and overlap with the theoretical framework, which incorporates the WHO health system building blocks (leadership and governance, service delivery, health system financing, health workforce, medical products, vaccines and technologies, and health information systems) [14,15,16]. This framework allows for a comprehensive analysis of the associations and relationships between the resources and components of a health system [14,15,16]. This approach considers the perspectives of patients and HCPs and allows for the identification of how resource inputs may influence outcomes in another [14,15,16]. There is a significant need for an increase in political governance regarding cervical cancer screening services. The results of our study further inform the conceptual framework. Women at risk and HCPs indicate the importance of education (input) and informed knowledge on cervical cancer and risk factors (desired outcome) to increase the uptake of cervical cancer screening in Tanzania.

### 5.1. Strategies for Increasing Knowledge of the Role of Screening and Preventive Care/Services

HCPs interviewed in our study emphasized the importance of education in increasing knowledge about the importance of cervical cancer screening and treatment services among women at risk, especially in rural settings. They also highlighted the importance of informed decision-making. Many HCPs agreed that providing women with the knowledge surrounding cervical cancer screening would increase their knowledge of cervical cancer and the importance of screening and preventive treatment, which may lead to an increase in screening rates. This is congruent with the findings of a 2012 study that evaluated determinants of acceptance of cervical cancer screening in Dar es Salaam, Tanzania [24]. Women who had attended at least primary school were more likely to attend screening in comparison with women who had never attended school [24]. The education gap is also evident in the reasons for which women present at the clinic [24]. One study found that of the women presenting for screening at ORCI, 90.7% had at least some education compared to only 43.1% of women presenting for treatment [25].

Health promotion plays a significant role in increasing the uptake of cervical cancer screening and treatment services among women in Tanzania. It is of particular importance to establish a greater number of programs that educate a large number of individuals in close vicinity to where they reside. This is to optimize convenience and reduce barriers such as transportation and distance. This is evident in studies, which show that differences in knowledge have been found between rural and urban areas in Tanzania, with generally lower knowledge and greater issues accessing cancer screening services in remote areas [25]. Other studies have found that women who are least likely to be aware of cervical cancer are rural women, less affluent women, those with limited education, and those with limited access to the formal economy [10]. In the aforementioned study conducted at the OCRI, women in the treatment clinic had to travel significantly further at a much greater cost than women in the screening clinic [25]. In another study, women having to come a long distance before reaching centers for cervical cancer screening was seen as a barrier by 58.7% of participants [26]. An example of a successful intervention for health promotion in Tanzania is, in fact, OCRI, which began a mobile screening clinic in 2001 and travels to various satellite clinics throughout the country [25]. This mobile clinic screens approximately 1500 women during the 2-week outreach session and trains HCPs to continue screening services upon completion of the outreach program [25]. Above all, it has shown to be a cost-effective method in successfully reaching rural populations and providing education to thousands of women who would otherwise not receive appropriate care [25].

Word-of-mouth and radio were identified as successful techniques in raising awareness about the existence of screening services amongst the population. This strongly aligns with research from high- and low-health-resource countries where strategies that deliver health promotion messaging via word of mouth and audio-visual channels are effective in making women more informed [27]. In contrast, research undertaken while implementing a human papillomavirus video education program for women participating in large cervical cancer screening campaigns in Tanzania found that education provided to women via word of mouth resulted in the inconsistent spreading of information among women and inefficient use of HCPs time [28]. It was enforced that educational interventions must be appropriately designed for the specific population of interest to see long-lasting changes in health-seeking behavior [28]. However, there is uncertainty among these findings: another study found that the influence of family and friends was found to be the second largest source of information and that friends and family were the most trustworthy stakeholders in improving knowledge, a concept that was reinforced in a study evaluating health promotion strategies for HPV vaccination for prevention of cervical cancer [29]. As for the use of radio in health promotion, a 2013 study found that compared with women who did not attend the screening service, women who attended listened regularly to the radio (OR 24.76; 95% CI, 11.49–53.33) [30]. Another study conducted in Zanzibar, Tanzania found that integration of health awareness themes into popular television and radio dramas might be effective in such promotion and prevention [31].

Our research identified fear as a primary reason for the refusal of screening among participants interviewed. In the aforementioned study conducted in Zanzibar, Tanzania, fear of screening and inconvenience were the primary concerns among the Zanzibari interviewees [31]. Another study conducted in Addis Ababa, Ethiopia, also found that perceived fear of positive results and the perceived pain and discomfort experienced with undergoing screening were mentioned as a barrier to seeking screening services [32]. A Kenyan study found that female participants stated fear and shame, or embarrassment related to the screening procedure as a barrier [33]. Specifically, participants were worried about being fully or partially nude during screening, especially with a male doctor [33]. To combat this, participants suggest that women should be screened where possible by a female health provider for screening to be more acceptable to women, although they may still fear being “judged” by the physician [33].

### 5.2. Education

Our study found that the level of training and breadth of knowledge among HCPs was felt to be integral to increasing rates of health promotion and uptake of screening services among women at risk. This is supported by a study that found that women undergoing cervical cancer screening at a health clinic reported that they had heard about cervical cancer screening from a HCP [34]. This is congruent with other studies, which have shown that HCP recommendations may positively influence the uptake of cervical cancer screening [35]. A cross-sectional study conducted in urban Tanzania found that those who received information about cervical cancer from HCPs were 1.569 times more likely to have adequate knowledge of cervical cancer as compared to respondents who did not receive cervical cancer information [11]. If women at risk are seeking knowledge from HCPs to ensure informed decision-making, it is integral that HCPs (doctors and nurses) are knowledgeable and resourceful. HCPs are pivotal in encouraging women to adopt protective health behaviors and health-seeking behaviors, correcting misconceptions about screening tests, and providing targeted comprehensive health education [35]. Our study is not the first to document the importance of knowledge and screening skills/competence among health providers as a barrier to cancer screening [35]. A 2018 study addressing barriers and facilitators to screening found that only approximately 70% of clinicians had performed cervical cancer screenings [36]. Of those who did receive training, most only received a 2 h orientation, which took place 2 to 5 years prior [36]. Participants were primarily trained to do Pap smears, VIA, and visual inspection with Lugol’s iodine (VILI) [36]. As a result, most participants felt that the training they received in screening and treating precancerous lesions was insufficient and contributed to a lack of confidence in performing screening procedures and preventive treatment [36].

Knowledge levels among women at risk were found to be an integral factor regarding cervical cancer screening uptake. This aligns with the results of similar studies conducted in Tanzania, which found that screening acceptance is associated with knowing about cervical cancer, its risk factors, and the role of prevention [24]. The need for education regarding cervical cancer prevention exists across sub-Saharan Africa, as several studies have found that women who lack knowledge of cervical cancer and the role of screening (especially in rural locations) are less likely to participate in screening services and are at greater risk for developing invasive cervical cancer [24].

The provision of health education with the end goal of disseminating information to male partners was highlighted as a topic of importance by one woman. One study highlighted the importance of male involvement in the cervical cancer cascade, as it found that, of the women presenting for screening, 43.6% came alone, and 12.8% were accompanied by at least one male relative [25]. Opposingly, in the treatment group, 16.3% came alone, and 44.9% were accompanied by a male [25]. Thus, it is evident that men play a role in their partner’s reproductive health experiences in many ways: shared decision-making, granting permission services, or providing financial support and transport for health services [37].

Knowledge levels before screening also significantly affected women’s screening acceptance in other studies. It was discovered that women who had attended at least primary school were more likely to attend screening than women who had never attended school [38]. It is thus evident that more work is required to address gaps in patient education at an earlier age as this will lead to better acceptance of cervical cancer screening services in the future.

### 5.3. Beliefs on Cervical Cancer Screening

Our research found that women believed that cervical cancer was caused by a variety of causes including contraceptives/medications, food, fate, and karma. There continues to be a lack of adequate knowledge surrounding cervical cancer among Tanzanian women and HCPs, serving as a barrier to cervical cancer screening. During a Prevention and Awareness Campaign in Northern Tanzania (2017–2019), a series of multiple-choice questions were distributed to participants to identify their beliefs and misbeliefs about the etiology and causes of cervical cancer [39]. A concerning number of women attributed the cause to using contraceptives (20%), while others selected it to be a result of a curse (14%), receiving the vaccine (5.8%), or through direct contact with a cancer patient (2.8%) [39]. Interestingly, incorrect responses were positively correlated with women living in rural regions of Tanzania [39].

In another study, a community-based cross-sectional was performed to gather predictors of cervical cancer screening [11]. A striking 1013 (91.8%) of the respondents had not been screened for cervical cancer [11]. Notably, screening intention, health beliefs, and knowledge level were identified as predictors for cervical cancer screening [11]. Interestingly, 53% had inadequate knowledge of cervical cancer and screening, 54.4% had no screening intention, 50% had negative health beliefs, and 53% had inadequate knowledge of cervical cancer and screening [11]. It was determined that respondents who had no intention to screen were less likely to uptake cervical cancer screening [11]. Respondents suffered many gaps surrounding the knowledge about VIA, especially in terms of its purpose, suggested age for screening, and the appropriate screening intervals [11]. There is a clear need to conduct community-awareness-raising campaigns and knowledge-raising campaigns to increase cervical cancer knowledge and establish peer-supporting screening programs in communities. It might be beneficial to explore peer-led navigation to promote cervical cancer screening knowledge, intention, and practices among urban women in Tanzania. Specifically, it is a form of task sharing that delegates cervical cancer-related tasks from HCPs to community health workers.

The lack of knowledge surrounding cervical cancer extends across various levels of Tanzania’s health system. Specifically, there continues to be a lack of education even among HCPs. In one study, it was identified that less than half of the nurses had adequate knowledge regarding cervical cancer [40]. Specifically, knowledge levels of causes of cervical cancer and transmission of HPV and age demonstrated a significant correlation [40]. Interestingly, levels of knowledge ranged across the levels of HCPs, although seemed to be more adequate among young nurses [40]. In terms of screening intervals being aware of the HPV vaccine, it was identified that registered nurses had more knowledge than enrolled nurses [40]. Notably, 84.6% of the nurses had never had a Pap smear examination [40]. Moreover, the results of this study indicate the pressing need to continue medical education and awareness across all levels of Tanzania’s healthcare sector.

### 5.4. Traditional Medicine

Reasons for seeking traditional medicine treatment may be an amalgamation of beliefs and financial strains, causing women to use local herbs rather than seek appropriate treatment for signs and symptoms that may be due to cervical cancer. A study of 655 adults in Northern Tanzania found that a majority of respondents use traditional medicines due to lower cost, fewer side effects, ease of access, and perceived safety and efficacy [41]. Importantly, it is not the inadequacy of the modern health system alone that influences women to seek traditional medicine, but rather a combination of factors related to cultural beliefs, past experiences, financial considerations, and potentially residing in a more rural location vs. urban where the use of traditional healers may be less prevalent and health literacy may be higher, especially with respect to screening services [41]. The use of traditional healers, particularly when signs and symptoms may already be present (and diagnosis is needed instead of screening) may be influenced by external factors such as fear of a cancer diagnosis and fear of the financial implications this may have on the family [41]. In some cases, traditional healers may deliberately promote the use of traditional medicines and their effectiveness in curing cancers [42]. This belief was the same in rural and urban sites as per a 2014 study conducted in northern Uganda [42].

### 5.5. Risk Factors/Symptoms and Signs

The HCPs interviewed in our study understood that HIV is a significant risk factor for cervical cancer. In one study, approximately four in ten Tanzanian women denied that they knew anything about the warning signs and risk factors for cervical cancer [31]. Many women commonly believed that oral contraceptives, condom usage, and swimming in public pools were risk factors [11]. Unfortunately, in a similar study, it was also identified that a high proportion of women were also completely ignorant about warning signs and risk factors [11]. Interestingly, 83.7% of respondents were not aware that family history is a notable risk factor for cervical cancer [11]. In addition, 83% and 78.4% were not aware that HIV/AIDS and pregnancies and deliveries, respectively, could increase the risk of disease [11].

In a community-based study, many women (68.3%) were not able to identify bleeding and spotting after menopause as signs of cervical cancer, and (72.1%) were not able to identify bleeding and feeling pain after intercourse, as possible signs of cervical cancer [11]. In another community-based cross-sectional study in the Ilala municipality in Tanzania, a total of 210 women were interviewed [43]. Among all responses, irregular vaginal bleeding was the most common symptom that was mentioned by participants (51.7%) [43].

## 6. Strengths and Limitations

One strength of this study is its inclusion of perspectives from different domains of healthcare (key informants, women at risk, and HCPs). This provides us with a comprehensive understanding of awareness and knowledge surrounding cervical cancer screening in rural Tanzania, as compared to other studies, which limit their findings to solely women in the community [31].

Another strength of the study was that semi-structured interviews were conducted to allow participants to better express their opinions and experiences in an open and free manner. This flexible interview protocol consists of open dialogue between the researcher and participant and encourages deeper conversations about personal and sensitive issues [44]. Additionally, the incorporation of inter-coder reliability offers several benefits, such as improving the systematicity, communicability, and transparency of the coding process: promoting reflexivity and dialogue within the research team and strengthening the reliability of the analysis [45].

A significant limitation of this study is its qualitative nature, resulting in an increased risk of error and bias and the potential for less internal validity.

Another limitation of this study is that we sampled participants from eight districts in four regions in Tanzania. The results should be more generalizable to semi-rural and rural health facilities and communities. The sampling should be expanded to include a more equal distribution of urban and rural participants.

## 7. Conclusions

There is a low level of knowledge of cervical cancer and the role of screening and prevention services among HCPs and women at risk in rural and semi-rural Tanzania. There is a critical need to implement strategies among health providers, within health facilities, in health promotion, and in rural communities to increase screening uptake and access to screening services among women at risk.

## Figures and Tables

**Figure 1 ijerph-21-01059-f001:**
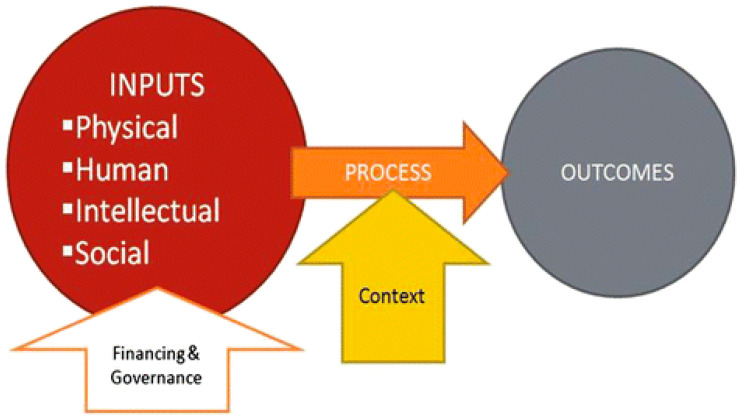
HSA Conceptual Framework Developed by Risso-Gill and colleagues for hypertension control (adapted from WHO Health Systems Building Blocks): (1) delivery of healthcare, (2) monitoring, (3) healthcare quality, access, and evaluation, (4) healthcare evaluation mechanisms, (5) national health programs—organization and administration [16].

**Figure 2 ijerph-21-01059-f002:**
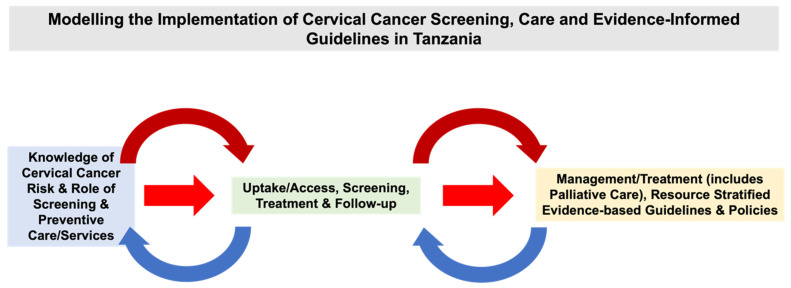
Modeling the Implementation of Cervical Cancer Screening, Care and Evidence-Informed Guidelines in Tanzania.

**Figure 3 ijerph-21-01059-f003:**
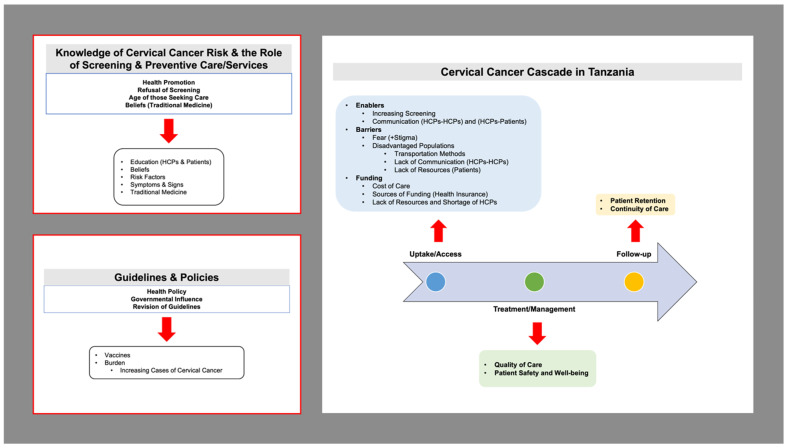
Knowledge of Cervical Cancer and the Role of Screening and Preventive Care/Services, the Cervical Cancer Cascade, and Guidelines and Policies for Cervical Screening in Tanzania.

**Table 1 ijerph-21-01059-t001:** List of Interviewed Participants (e.g., Location, Number Interviewed Per Group, Median Age). Assistant Medical Officer—AMO; Clinical Officer—CO; Medical Doctor—MD; Medical Officer—MO; Regional Reproductive and Child Health Coordinator—RRCHCo.

Interview Participants					
Group	Location	No. Interviewed	Median Age	No. Females	No. Males
Cervical cancer screening program leadership	KCMC, ORCI, Bombo Referral Hospital	8 (1 = Senior Lecturer and Senior Consultant for Gynecology and Obstetrics, 2 = Specialist (Oncology), 1 = Professor Gynecology and Obstetrics, 1 = Gynecologist, 1 = Cervical Cancer Focal Person MOH, 1 = RRCHCo, 1 = In Charge of Clinic	Not collected	4	4
Healthcare Providers	Mt Meru, Marangu Hospital, Meru District Hospital, Pasua Health Center, TPC Hospital, Kibosho Hospital, ORCI, Kilema, SEVIA Screening Sites, KCMC, Bombo Referral Hospital	24 (AMO = 2, MD = 12, MO = 1, Nurse = 9)	Not collected	13	11
Women at Risk of Cervical Cancer (Women at Risk/Clients)	ORCI, Kilema, TPC, Pamoja Tunaweza Women’s Centre, SEVIA Screening Sites	39	46 (18–80 years old, 50 years old as the most common age, three unknown ages)	39	0

**Table 2 ijerph-21-01059-t002:** Health Facilities Sampled by Site Characteristics (Rural/Semi-Rural vs. Urban; Private vs. Public vs. Faith-Based).

Location	Rural/Urban	Private/Public/Faith-Based
Kilimanjaro Christian Medical Center (KCMC)	Urban	Faith-Based/Public Hybrid
Mt. Meru Hospital	Urban	Public
Ocean Road Cancer Institute (ORCI)	Urban	Public
Kilema Hospital	Rural	Faith-Based
TPC Hospital	Rural	Private
Marangu Hospital	Rural	Faith-Based
Meru District Hospital	Urban	Public
Pasua Health Center	Urban	Public
Kibosho Hospital	Rural	Faith-Based
Bombo Regional Referral Hospital	Semi-Rural	Public

**Table 3 ijerph-21-01059-t003:** Seven Identified Themes and Examples of Evidence.

Themes	Sub-Themes	Examples of Evidence
Knowledge of the role of screening and preventive care/services	Health Promotion, Word of Mouth, Radio, and Refusal of Screening	“Awareness should be spread in different areas…Most people don’t know anything about cervical cancer. So, awareness should be provided”. (Patient)
Knowledge and Training (HCPs)	Training	“I don’t think even among the health profession… that people know exactly how to walk through it (HCP)”.
Knowledge (Women at Risk)	-	“Provision of education is needed... [healthcare providers] are supposed to [provide education]. [If we can educate men], a lot of women would be willing to go for screening (HCP)”.
Beliefs on Cervical Cancer Screening	-	“She said that contraception can cause cervical cancer” (Patient)
Traditional Medicine	-	“They want to go to a traditional attendant or a traditional doctor who can say, ‘This is just someone [who has] bewitched you. I can take care of you’“ (HCP)
Risk Factors	HIV, Early Sexual Intercourse	“They mentioned.. that [those who are] HIV-positive are at risk of getting cervical cancer” (Patient)
Symptoms and Signs	Pain	“[Elderly] women come and complain about... bleeding, post-bleeding and... discharge. When you insert the speculum... there is cancer” (HCP)

## Data Availability

The data presented in this study are available on request from the corresponding author due to privacy and ethical reasons.

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
