# Peer review of "“In the Village That She Comes from, Most of the People Don’t Know Anything about Cervical Cancer”: A Health Systems Appraisal of Cervical Cancer Prevention Services in Tanzania"

_ijerph, 2024, doi:10.3390/ijerph21081059_

Round 1
Reviewer 1 Report
Comments and Suggestions for Authors
The authors conducted a qualitative study to better understand the perception, knowledge, and barriers to cervical cancer screening. The study is interesting and relevant and would be important to healthcare providers, policymakers, and the population of Tanzania. I have provided some comments and suggestions to improve the manuscript. Better organization of the text would facilitate understanding.
Abstract:
I recommend breaking the first sentence into two sentences. The first sentence could end at “…in women globally.”
At the end of the Introduction, state the main objective(s) of the study.
Always use the verb ‘were’ after data.
In Conclusion, start the sentence with “Our results show…” instead of “It is evident…”
Introduction:
I recommend shortening the length of this section; the 1st paragraph on page 2 could be summarized.
The last paragraph seems more appropriate in the Methods section.
Place the details of the objectives of the study as the last paragraph.
Methods:
I feel that there is too much information on the HAS framework; suggest shortening and letting readers know that additional details can be found in the references.
Is Figure 1 necessary in this paper?
Provide more details on the inclusion and exclusion criteria for the sample.
State the number of sites visited, the number of participants at each site and number who agreed to participate. Then provide details on the teams that conducted the interviews, etc.
Data collection:
It would be helpful to readers to provide the interview guide in Supplementary tables.
How did you determine “women at risk of cervical cancer”?
Since open-ended questions were used, it would be useful to see those questions. You can place these in a Supplementary table.
Data analysis:
Re: “Statement related to physical, human, intellectual, and social resources were coded…” Provide the statement in a Supplemental table and describe how the coding was done.
Results:
Last sentence on page 8: How are the last two themes different from the 1st theme? Do you think the themes should be reworded?
Table 3: It might be helpful to include the sub-themes inside this table.
Just a suggestion on the formatting. Since you bolded the 1st theme, it would be helpful to also bold each theme (but not the sub-topics).
Discussion:
I recommend starting this section by restating the objectives of your study. Then discuss your results before presenting the literature review of articles that supported or did not support your results. This was done in a few sections but not all.
Strengths and limitations:
The last sentence of this section is not quite clear to me. As written, it doesn’t read as a limitation.
Comments on the Quality of English LanguageMinor English language editing is needed.
Author Response
Please see the responses to Reviewer 1's comments attached.

Reviewer 2 Report
Comments and Suggestions for Authors
Very interesting paper
Only some suggestions
It is certainly very important to raise patient awareness but the importance of treating patients appropriately should not be underestimated
Particulary, doctors adequately trained in external beam radiotherapy and brachytherapy (Is an Interventionaò Oncology Center an advantage in the service of cancer patients or in the rducation? The Gemelli Hospital and INTERACTS experience; Quality assurance in modern gynecological hdr‐brachytherapy (Interventional radiotherapy); clinical considerations and comments)
Cervix cancer patients are long-survivors, it is mandatory to preserve the quality of life (Vaginal dilator use more than 9 months is a main prognostic factor for reducing G2‑late vaginal complications in 3D‑vaginal‑cuff brachytherapy (interventional radiotherapy)?Clinical and Translational Oncology 2023; 25(6), pp. 1748-1755
Vaginal Toxicity Management in Patients with Locally Advanced Cervical Cancer following Exclusive Chemoradiation—A Nationwide Survey on Knowledge and Attitudes by the Italian Association of Radiotherapy and Clinical Oncology (AIRO) Gynecology Study Group
The impact of Happy (Humanity Assurance Protocol in Interventional Radiotherapy) on the psychological well-being of gynecological cancer patients)
It have to be mandatory the discussion of patients in multidisciplinary team (Can A Dedicated Multidisciplinary Tumor Board Improve Personalized Medicine for Patients in Interventional Oncology? A Large Retrospective Single-Center Experience)
Author Response
Please see the responses to Reviewer 2's comments attached.

Reviewer 3 Report
Comments and Suggestions for Authors
In this article, the authors adapted Risso-Gill and colleagues’ framework for a HSA, to identify HCPs' perspective of the extent to which health system requirements for effective cervical cancer screening, prevention, and control are in place in Tanzania. The manuscript is straightforward, well written, and concise. Definitely deserves to be published and is a valuable contribution to the “International Journal of Environmental Research and Public Health”. The following comments need to be addressed before publication, as recommended.
[1] “Introduction”, Pages 1 of 20 and 2 of 20:
“While there has been a reduction in cervical cancer prevalence rates since the introduction of Pap smears in the mid-20th century, healthcare systems and care vary across the globe [2].”.
At that point, the authors should report that NHS England first aimed to tackle the burden of the disease by introducing a national cervical screening programme in 1988, which has since seen a significant reduction in over a third of cases in England. Cervical cancer screening is available from the age of 25, as the disease is rare among younger individuals.
Recommended reference: Choi S, et al. HPV and Cervical Cancer: A Review of Epidemiology and Screening Uptake in the UK. Pathogens 2023;12(2):298.
[2] “Introduction”, Page 2 of 20:
“Presently, the WHO recommends using HPV DNA sampling as the primary screening test as opposed to visual inspection with/through acetic acid (VIA) or cytology for screening and treatment purposes among both the general population, as well as women living with HIV [4].”.
At that stage, the authors should mention that from the therapeutic perspective, there is increasing interest in the role of immunotherapy in advanced cervical cancer, particularly as the causative role of HPV is well established. A number of immunotherapy trials have been undertaken evaluating vaccine-based therapies, adoptive T-cell therapy and immune-modulating agents in patients with advanced cervical cancer.
Recommended reference: Grau-Bejar JF, et al. Advances in immunotherapy for cervical cancer. Ther Adv Med Oncol. 2023;15:17588359231163836.
Minor editing of English language required.
Author Response
Please see the responses to Reviewer 3's comments attached.

Round 2
Reviewer 1 Report
Comments and Suggestions for Authors
The authors have addressed comments to my satisfaction. I find that the revised version is now much improved. I have no additional comments.